# Variant Analysis of the Thymidine Kinase and DNA Polymerase Genes of Herpes Simplex Virus in Korea: Frequency of Acyclovir Resistance Mutations

**DOI:** 10.3390/v15081709

**Published:** 2023-08-09

**Authors:** Jungwon Hyun, Su Kyung Lee, Ji Hyun Kim, Eun-Jung Cho, Han-Sung Kim, Jae-Seok Kim, Wonkeun Song, Hyun Soo Kim

**Affiliations:** 1Department of Laboratory Medicine, Hallym University Dongtan Sacred Heart Hospital, Hallym University College of Medicine, Hwaseong 18450, Republic of Korea; jungwonhyun@hallym.or.kr (J.H.); sklee1217@naver.com (S.K.L.); wlgusrob@naver.com (J.H.K.); ejlovi@hallym.or.kr (E.-J.C.); 2Department of Laboratory Medicine, Hallym University Sacred Heart Hospital, Hallym University College of Medicine, Anyang 14068, Republic of Korea; kimhs@hallym.or.kr; 3Department of Laboratory Medicine, Kangdong Sacred Heart Hospital, Seoul 05355, Republic of Korea; jaeseokcp@gmail.com; 4Department of Laboratory Medicine, Hallym University Kangnam Sacred Heart Hospital, Hallym University College of Medicine, Seoul 07441, Republic of Korea; swonkeun@hallym.or.kr

**Keywords:** HSV-1, HSV-2, antiviral resistance, variants, next-generation sequencing, polymorphisms

## Abstract

The thymidine kinase (TK) and DNA polymerase (pol) genes of the herpes simplex viruses type 1 (HSV-1) and type 2 (HSV-2) are two important genes involved in antiviral resistance. We investigated the genetic polymorphisms of the HSV-TK and pol genes in clinical isolates from Korean HSV-infected patients using next-generation sequencing (NGS) for the first time in Korea. A total of 81 HSV-1 and 47 HSV-2 isolates were examined. NGS was used to amplify and sequence the TK and pol genes. Among the 81 HSV-1 isolates, 12 and 17 natural polymorphisms and 9 and 23 polymorphisms of unknown significance in TK and pol were found, respectively. Two HSV-1 isolates (2.5%) exhibited the E257K amino acid substitution in TK, associated with antiviral resistance. Out of 47 HSV-2 isolates, 8 natural polymorphisms were identified in TK, and 9 in pol, with 13 polymorphisms of unknown significance in TK and 10 in pol. No known resistance-related mutations were observed in HSV-2. These findings contribute to our understanding of the genetic variants associated with antiviral resistance in HSV-1 and HSV-2 in Korea, with frequencies of known antiviral resistance-related mutations of 2.5% and 0% in HSV-1 and HSV-2, respectively.

## 1. Introduction

Herpes simplex viruses type 1 (HSV-1) and type 2 (HSV-2) are double-stranded DNA viruses of approximately 150 kbp in size that cause orolabial and genital skin infection, keratitis, encephalitis, and neonatal disease [1,2]. Acyclovir and penciclovir, with their respective prodrugs valacyclovir and famciclovir targeting DNA polymerase, are first-line antiviral agents for HSV infection treatment [1]. Both acyclovir and penciclovir are selectively phosphorylated by HSV thymidine kinase (TK) and are then further phosphorylated by cellular kinases, which leads to acyclovir or penciclovir triphosphate production [1]. These triphosphate forms compete with deoxynucleotide triphosphates (dNTP), and the incorporation of these analog triphosphates into the growing DNA chain terminates HSV DNA replication [1]. Therefore, mutations in HSV-TK and HSV-DNA polymerases inhibit HSV replication. Mutations in the HSV-TK and HSV-DNA polymerase genes (*UL23* and *UL30*, respectively) confer HSV acyclovir resistance, with HSV-TK mutations accounting for 95% of resistance [1]. Several HSV-TK and DNA polymerase mutations have been reported, some of which are natural polymorphisms, while others are related to acyclovir resistance [3].

The prevalence of acyclovir-resistant HSV-1 and HSV-2 has been reported to be as low as 0.1–0.7% in immunocompetent individuals in most studies, but a report documented a relatively high prevalence (6.4%) in immunocompetent patients with herpetic keratitis [4,5]. However, in immunocompromised patients, the prevalence of HSV-1 and HSV-2 varies from 3.5 to 10% and is up to 36% (5 of 14 patients) in patients who underwent hematopoietic stem cell transplant [5,6]. Therefore, the emergence of acyclovir resistance is an important clinical problem that poses a barrier to HSV infection treatment, especially in immunocompromised patients. Although clinically relevant, there have been no reports on natural polymorphisms or resistance-related nucleotide changes in HSV-TK (*UL23*) and DNA polymerase (*UL30*) genes in Korea. In this study, we aimed to determine the variation frequencies of the TK and DNA polymerase genes of HSV-1 and HSV-2 isolated from Korean patients using next-generation sequencing (NGS).

## 2. Materials and Methods

### 2.1. Study Participants and Clinical Samples

Between May 2017 and March 2018, 81 HSV-1- and 47 HSV-2-positive residual samples after testing for HSV by PCR or culture were collected from patients who visited the Hallym University Dongtan Sacred Heart Hospital in Korea. The samples were stored at −70 °C. The patient clinical data were obtained from medical records. HSV-1 samples (*n* = 81) were collected from 30 male (37.0%) and 51 female (63.0%) patients, with an overall median age of 35 years (range, 1–88 years) (Table 1). HSV-2 samples (n = 47) were collected from 17 male (36.2%) and 30 female (63.8%) patients, with an overall median age of 50 years (range, 20–82 years) (Table 1). A large majority of the obtained specimens were cutaneous skin swabs (n = 58 [71.6%] for HSV-1 and n = 27 [57.4%] for HSV-2). Of the 128 patients, 101 were documented in the medical records to have used antivirals (acyclovir, famciclovir, or valaciclovir) for 1 to 12 days after HSV diagnosis. This study was approved by the Institutional Review Board of Hallym University Dongtan Sacred Heart Hospital (IRB No. NON2018-003, 2022-02-007).

### 2.2. UL23 and UL30 Gene PCR and Sequencing of HSV-1 and HSV-2 Using NGS

Viral DNA extraction from HSV-positive samples and culture media was performed using the QIAamp DNA mini kit (Qiagen, Hilden, Germany) and QIAcube platform (Qiagen) according to the manufacturer’s instructions. The HSV-TK and DNA polymerase genes were amplified from the extracted DNA samples using gene-specific primer sets (Table 2) and 2.5 U of AmpliTaq Gold DNA polymerase (ThermoFisher Scientific, Waltham, MA, USA). The PCR conditions were as follows. For HSV-1 TK and Pol 1 and Pol 2, initial denaturation at 95 °C for 5 min was followed by 35 cycles at 95 °C for 1 min, 55 °C for 1 min, 72 °C for 90 s and final extension at 72 °C for 7 min. For HSV-2 TK, initial denaturation at 95 °C for 5 min was followed by 35 cycles at 95 °C for 1 min, 55 °C for 1 min, 72 °C for 90 s and final extension at 72 °C for 7 min. For HSV-2 pol 1, initial denaturation at 95 °C for 5 min was followed by 35 cycles of 95 °C for 1 min, 60 °C for 1 min, 72 °C for 90 s and final extension at 72 °C for 7 min. For HSV-2 pol 2 and pol 3, initial denaturation at 95 °C for 5 min was followed by 35 cycles of 95 °C for 1 min, 52 °C for 30 sec, 72 °C for 90 s and final extension at 72 °C for 7 min.

To create 300 bp fragment libraries, the PCR products were pooled in equimolar amounts, fragmented, and subsequently ligated to barcoded adaptors using an Ion Xpress Plus Fragment Library Kit (Thermo Fisher Scientific, Waltham, MA, USA) and Ion Express Barcode Adapter kits (Thermo Fisher Scientific), respectively. Template preparation, including emulsion PCR, was performed using an Ion 510 & Ion 520 & Ion 530 kit-Chef (Thermo Fisher Scientific) and an Ion Chef system (Thermo Fisher Scientific). NGS was performed using the Ion Torrent S5 XL NGS platform (Thermo Fisher Scientific) and the Ion S5 sequencing kit (Thermo Fisher Scientific) on a 520 chip (Thermo Fisher Scientific). The sequence reads were quality-trimmed using Ion Torrent Suite version 5.0.4 (Thermo Fisher Scientific). The raw sequence data were analyzed using the CLC Genomics Workbench version 11 (Qiagen, Aarhus, Denmark). The sequence reads were trimmed and mapped to the HSV reference sequences (NC_001806.2 for HSV-1 and NC_001798.2 for HSV-2), and variation analysis of HSV-TK and DNA polymerase genes was performed. The obtained sequences were analyzed and aligned using the Molecular Evolutionary Genetics Analysis (MEGA) program (version 7).

### 2.3. Variation Analysis of the UL23 and UL30 Genes in HSV-1 and HSV-2

The obtained sequence variations were analyzed, and the frequencies of each variation were calculated. To compare the variation frequency in the HSV-TK and DNA polymerase genes between this study and previous studies [7,8,9], the Chi-square and Fisher’s exact tests were performed using MedCalc software, version 19.8 (MedCalc Software Ltd., Ostend, Belgium); *p* < 0.05 was considered statistically significant.

### 2.4. Phylogenetic Analysis of the UL23 and UL30 Genes in HSV-1 and HSV-2

Phylogenetic analyses were performed to determine the relationship between the deduced amino acid sequences of the HSV-TK and HSV-DNA polymerase genes in this study and the previously reported amino acid sequences corresponding to the HSV-TK and DNA polymerase genes. The previously reported amino acid sequences were obtained from NCBI virus (ncbi.nlm.nih.gov/labs/virus/) accessed on 16 March 2023. A total of 879 and 212 complete protein sequences for HSV-1 *UL23* and *UL30*, respectively, and 316 and 79 complete protein sequences for HSV-2 *UL23* and *UL30*, respectively, were downloaded. Since there were too many sequences to draw a phylogenetic tree on one page, we drew phylogenetic trees with our research results and reference sequences (NC_001806.2 for HSV-1 and NC_001798.2 for HSV-2). The deduced amino acid sequences were aligned, phylogenetic trees were obtained using MEGA software, and the amino acids different from the reference sequences are shown.

## 3. Results

### 3.1. Polymorphisms of the Thymidine Kinase (UL23) Genes in HSV-1

Polymorphisms in the HSV-TK gene were investigated among 81 HSV-1 clinical isolates. Compared to the HSV-1 reference sequence (NCBI Reference sequence: NC_001806, YP 009137097, collected in the United Kingdom before 1972), the number of nucleotide substitutions in UL23 in our study ranged between 8 and 12 (mean = 9.8) among a total of 1128 bp, and the number of deduced amino acid substitutions in UL23 ranged from 3 to 7 (mean = 4.9) among a total of 376 amino acids (Table 3). Compared with the reference sequence, the % nucleotide identity was >98% in all strains.

The amino acid substitution K36E was found in all isolates (Table 4, Figure 1), N23S and A265T in 80 (98.8%), L42P in 52 (64.2%), C6G in 49 (60.5%), R89Q in 28 (34.6%), V348I in 8 (9.9%), and E257K in 2 (2.5%) isolates. The amino acid substitutions Y4H, C6V, Q34P, R41H, D76E, E95Q, E146Q, S276R, G279D, A294V, D303N, R320C, D330N, G335S, and I361L were each found in one isolate (1.2%) (Table 4, Figure 1). The amino acid substitutions C6G, N23S, K36E, R41H, L42P, R89Q, E146G, A265T, S276R, A294V, D303N, and V348I are known natural polymorphisms, and E257K is known to confer acyclovir resistance [3]. Among the other 10 amino acid changes identified in this study, Y4H, C6V, Q34P, E95Q, R320C, D330N, and I361L are novel polymorphisms and have not been previously reported. We compared the frequencies of the variants identified in this study with those reported in previous studies [6,7,8]. The frequencies of C6G and L42P observed in the current study were significantly higher than those reported in previous studies (Table 4). In addition, the frequencies of K36E and A265T were significantly higher than those reported in a German study [7,8], while the frequency of R89Q was significantly lower than that reported in a French study [6]. Figure 1 shows the phylogenetic analysis and reveals that the HSV-1 UL23 amino acid sequences in our study differed by 3–7 amino acids from the reference sequence (NC_001806 [YP 009137907]); the HSV-1 UL23 sequences with C6G, N23S, K36E, L42P, and A265T variations were the most common (40/81, 49.4%).

### 3.2. Polymorphisms of the DNA Polymerase (UL30) Gene in HSV-1

The polymorphisms of the HSV-DNA polymerase gene were investigated among 81 HSV-1 clinical isolates (Table 3). When the total HSV-1 UL30 3705 nucleotide sequences and 1235 amino acid sequences were compared with the reference sequence HSV-1 (NC_001806.2, YP_009137105, collected prior to 1972 in the United Kingdom), the number of nucleotide substitutions in UL30 in our study ranged between 11 and 17 (mean of 13.3) among a total of 3705 bp, and the number of deduced amino acid substitutions in UL30 ranged between 4 and 8 (mean of 5.8) among a total of 1235 amino acids (Table 3). Compared with the reference sequence, the nucleotide identity% was >98% for all strains.

The amino acid change S33G was found in all isolates (Table 5), T1208A in 80 (98.8%); V905M in 78 (96.3%), P1124H in 76 (93.8%), P920S in 57 (70.4%), D672N in 47 (58.0%); P1199Q in 5 (6.2%), I182V, A1099T, and S1113C in 4 (4.9%), R112H, R663W, G749D, K908Q, and A1098T in 3 (3.7%) isolates, and P8T, A25V, P29H, A92T, D110A, P136T, E249K, E254A, R264H, V278A, E353K, A405V, D871N, P875T, E1104K, E1120K, P1198H, P1198L, and T1208V were each observed in 1 isolate (1.2%) (Table 5, Figure 2). The amino acid substitutions A25V, P29H, S33G, R112H, R264H, D672N, G749D, D871N, V905M, P920S, A1099T, E1104K, S1113C, P1124H, P1199Q, and T1208A are known natural polymorphisms. The other 16 amino acid changes identified in the current study are novel polymorphisms and have not been previously reported. The S33G, D672N, V905M, P920S, P1124H, and T1208A frequencies observed in this study were significantly higher than those reported in previous studies [6,7,8] (Table 5). No known resistance-related mutations were identified in the DNA polymerase gene in HSV-1 [3]. Figure 2 shows the phylogenetic analysis and reveals that the HSV-1 UL30 amino acid sequences in our study differed by 4–8 amino acids from the reference sequence (NC_001806 [YP_009137105]), and the HSV-1 UL30 sequences with S33G, D672N, V905M, P920S, P1124H, and T1208A variations were the most common (41/81, 50.6%) (Figure 2).

### 3.3. Polymorphisms of the Thymidine Kinase (UL23) Gene in HSV-2

The polymorphisms of the HSV-TK gene were investigated among 47 HSV-2 clinical isolates. When the total HSV-2 UL23 1128-nucleotide sequences and 376-amino acid sequences were compared with the reference sequence (NCBI Reference sequence: NC_001806, YP_009137174, collected prior to 1971 in the United Kingdom), the number of nucleotide substitutions in UL23 ranged between 0 and 6 (mean of 3.1) among a total of 1128 bp, and the number of deduced amino acid substitutions in UL23 ranged between 0 and 6 (mean of 2.8) among a total of 376 amino acids (Table 3). Compared to the reference sequence, the % nucleotide identity was >98% in all strains.

One isolate (#24) showed the same sequences as the reference sequences (Figure 3). The amino acid change G39E was found in 45 isolates (95.7%) (Table 6), N78D in 37 (78.7%), T159I in 10 (21.3%), A27T and S29A in 9 (19.1%), R26H and R220K in 5 (10.6%), L140F in 4 (8.5%), A215T in 3 (6.4%), and S38A in 2 (4.3%) isolates. Additionally, A219V, R294H, R294S, and I359V were each observed in one isolate (2.1%). The amino acid substitutions R26H, A27T, S29A, G39E, N78D, L140F, T159I, A215T, and R220K are known natural polymorphisms. The five newly identified amino acid changes are novel polymorphisms that have not been previously reported. The frequency of L140F observed in the present study was significantly lower than that reported in previous studies, and the frequency of N78D was significantly higher than that reported in a French study [6] (Table 6, Figure 3). No known resistance-related mutations were identified in the TK genes in HSV-2 [3]. Figure 3 shows the phylogenetic analysis and reveals that the HSV-2 UL23 amino acid sequences in our study differed by 0–6 amino acids from the reference sequence (NC_001798 [YP 009137174]), wherein the HSV-2 UL23 sequences with the G39E and N78D variations were the most common (15/47, 31.9%).

### 3.4. Polymorphisms of the DNA Polymerase (UL30) Gene in HSV-2

The polymorphisms of the HSV-DNA polymerase gene were investigated among 47 HSV-2 clinical isolates. When the total 3720-nucleotide HSV-2 UL30 sequences and 1240-amino acid sequences were compared with the reference sequence (NCBI Reference sequence: NC_001806, YP_009137182, collected prior to 1971 in the United Kingdom), the number of nucleotide substitutions in UL30 ranged from 3 to 11 (mean of 6.0) among a total of 3720 bp, and the number of deduced amino acid substitutions in UL30 ranged from 3 to 6 (mean of 3.8) among a total of 1240 amino acids (Table 3). Compared to the reference sequence, the % nucleotide identity was >99% for all strains.

The amino acid substitutions P15S and L60P were found in all isolates (Table 7, Figure 4), A9T in 42 (89.4%), E139K in 21 (44.7%), T801P in 6 (12.8%), and D676G as well as E678G in 2 (4.3%) isolates. The amino acid changes C40W, P138H, E191K, G684_G685del, D716A, Q792R, A1000T, I1033V, and A1065T were each observed in one isolate (2.1%). The amino acid substitutions A9T, P15S, C40W, L60P, E139K, D676G, E678G, T801P, and A1000T are known natural polymorphisms. The other seven amino acid changes observed in this study are novel polymorphisms that have not been previously reported. The frequency of amino acid changes observed in this study was not significantly different compared to those reported in previous studies (Table 7). No known resistance-related mutations were identified in the DNA polymerase genes of HSV-2 [3]. The phylogenetic analysis revealed that the HSV-2 UL30 amino acid sequences in our study differed by 3–6 amino acids from the reference sequence (NC_001798 [YP 009137182]), with the A9T, P15S, and L60P variations being the most common (17/47, 36.2%) (Figure 4).

## 4. Discussion

In this study, we identified the genetic variations of the TK and DNA polymerase (pol) genes and their frequencies among 81 HSV-1 and 47 HSV-2 strains in Korea. Compared with the reference sequences of HSV-1 and HSV-2, the fact that the % nucleotide identity was >98% in all strains in our study showed that the *UL23* and *UL30* genes were highly conserved. In general, the % nucleotide identity of viral genes is high between the same species collected in the same region during the same period; however, different mutations gradually occur over time. However, the fact that the nucleotide identity was >98% between the HSV-1 and HSV-2 strains detected in this study and the HSV-1 and HSV-2 reference strains detected before 1979 indicates that HSV-1 and HSV-2 are viruses that rarely have mutations in the *UL23* and *UL30* genes. In the case of rotavirus, a % nucleotide identity of <80% was demonstrated, even among the same genotype [10]. One reason for this phenomenon is that rotavirus is an RNA virus, while HSV is a double-stranded DNA virus [11]. However, a comparative study on the viral mutation rate of other dsDNA viruses is required because the viral mutation rate can be determined by other factors, such as proliferation rate, prevalence, use of antivirals, and host immune response [11]. The frequency of nucleotide and amino acid changes was higher in the TK gene than in the pol gene, and the mutation frequency was higher in HSV-1 than in HSV-2, similar to the findings reported in previous studies [7,8].

Few studies report the variation frequency of HSV-1 and HSV-2 genes and antiviral resistance. When we searched the PubMed database, there were only three studies that showed the frequency of each mutation so that the mutation frequencies obtained in our study could be compared (Table 3) [7,8,9]. Other studies reported HSV sequences but did not report the frequency of variation. Most variants found in our study with two or more frequencies were already known natural polymorphisms, while the E257K mutation was a known resistant-associated mutation. Several other new amino acid mutations were observed in our study. In 2015, Sauerbrei et al. reported natural polymorphisms and resistance-associated mutations for HSV mutations in a review article [3]. However, the frequency of each variation was unknown in this report; many of the resistance-associated mutations described in this report are not common across studies and have been reported in only one study with a frequency of 1–2. The E257K mutation, a resistance-associated mutation in our study, was reported in only one other study [9]. Therefore, further research on HSV mutation and antiviral resistance should be conducted to accumulate additional data on the relationship between each mutation and antiviral resistance.

Most mutations related to drug resistance in both TK and DNA polymerase genes are located within conserved regions [12]. The conserved gene regions II and III in the DNA polymerase contain >40% of resistance mutations [13]. In this study, 7 new amino acid changes in HSV-1 TK, 16 in HSV-1 pol, 5 in HSV-2 TK, and 8 in HSV-2 pol were identified. A large majority of the newly discovered variants were located outside the active or conserved gene regions [3,8]. However, one amino acid substitution, A219V, was found in the conserved domain of the HSV-2 TK gene; another amino acid substitution, Q792R, was located in the HSV-2 VI region of the DNA polymerase gene. Further studies are needed to determine whether these two variations are related to drug resistance.

HSV drug resistance testing can be performed by genotypic and phenotypic methods. The genotypic test uses TK and pol sequencing to identify known drug resistance mutations, as done in this study [14]. Natural polymorphisms and resistance-related mutations of the TK and pol genes from genotyping tests have been extensively described [3,15]. Variations of unknown significance are discovered in genotyping tests, and phenotyping tests can be performed to determine if the variations are related to drug resistance. When several unknown amino acid substitutions and resistance mutations occur within the TK and pol genes, determining their role is challenging. Usually, to determine the role of individual amino acid variations, recombinant viruses containing only one amino acid change can be generated using a set of overlapping cosmids and plasmids or the application of bacterial artificial chromosomes (BACs) [1,3,16]. The phenotypic test is based on virus culture in the presence of antiviral drugs; subsequently, resistance is determined by calculating the drugs’ inhibitory concentrations. Unfortunately, phenotypic tests for HSV resistance could not be performed in this study; therefore, whether the new mutations were related to drug resistance could not be confirmed. In this study, the known acyclovir resistance-related E257K mutation in the TK gene was found in 2 of the 81 HSV-1 samples, with a frequency of 2.5%. The acyclovir resistance rate in immunocompetent patients is <1% [17,18]. However, one study reported acyclovir resistance at 4.0% in immunocompetent pediatric patients [19] and up to 6.4% in immunocompetent patients with herpetic keratitis [5]. Overall, the acyclovir resistance rate of HSV in immunocompromised patients was higher and varied depending on the type of immunosuppression [19]. In patients with HIV, a high HSV acyclovir resistance rate of 3.4–7.3% was reported; an even higher rate of up to 46.5% was reported in patients who underwent hematopoietic stem cell transplant [17,20]. No immunosuppressed patients were included in the present study; the two patients with the E257K mutation were diagnosed with uncomplicated cutaneous HSV-1 infection without underlying immunosuppression. In this study, we did not observe acyclovir-resistant mutations among the HSV-2 clinical isolates. The frequency of acyclovir-resistant HSV-2 varies from study to study and ranges between 0 and 25% [14,20].

Our study has some limitations: (1) although immunocompromised individuals present a higher rate of drug resistance mutations, only six immunosuppressed patients or patients with malignancies were included in the present study; (2) we could not confirm whether the newly identified mutations were related to drug resistance. Therefore, further investigations on HSV drug resistance genes in various populations in Korea, including immunocompromised individuals, are required; (3) we did not perform additional Sanger sequencing to confirm the variants, although we checked the coverage, counts, and raw alignment data of all variants using the CLC Genomics Workbench. Therefore, caution should be taken when interpreting variants with frequencies of approximately 1%.

Nevertheless, this study provides specific nucleotide sequences for the TK and pol genes of HSV-1 and HSV-2, which can be used as important primary data for future research on HSV drug resistance-related genes in patients of various ethnicities, including Korean populations. In addition, the data in the current study can be used for future studies to determine the relationship between genetic variations and the therapeutic effect of antiviral drugs.

## 5. Conclusions

For the first time in Korea, we sequenced acyclovir resistance-related genes in the TK and pol genes of HSV-1 and HSV-2. We determined that the frequencies of known antiviral resistance-related mutations in HSV-1 and HSV-2 were 2.5% and 0%, respectively. The current data provide insight into the genetic variants of HSV within the Korean population and a basis for further antiviral treatment and resistance studies.

## Figures and Tables

**Figure 1 viruses-15-01709-f001:**
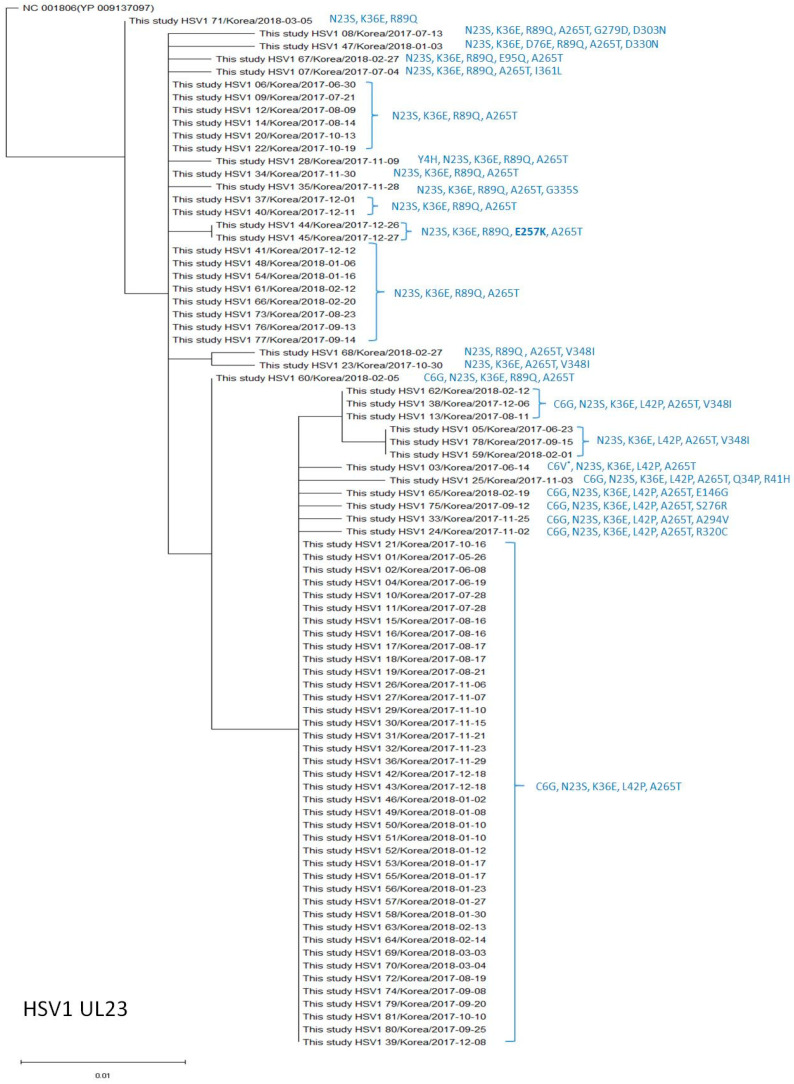
Phylogenetic analysis of the HSV-1 *UL23* amino acid sequences in this study and the reference sequence. * Amino acid substitutions different from the reference sequence are indicated on the right, next to the strain name.

**Figure 2 viruses-15-01709-f002:**
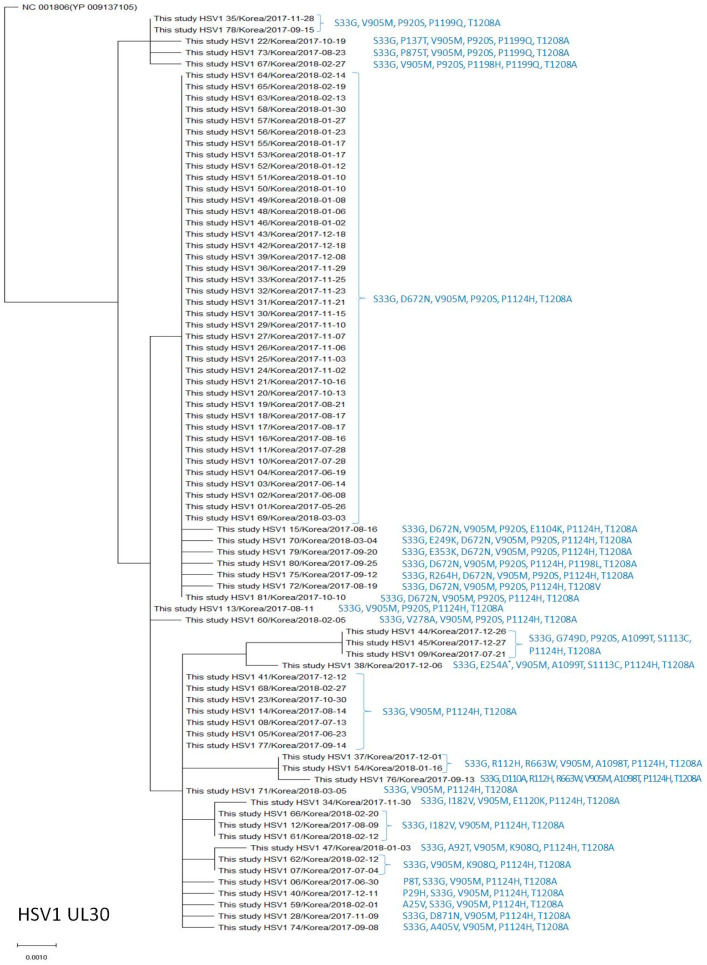
Phylogenetic analysis of the HSV-1 *UL30* amino acid sequences in this study and the reference sequence. * Amino acid substitutions different from the reference sequence are indicated on the right, next to the strain name.

**Figure 3 viruses-15-01709-f003:**
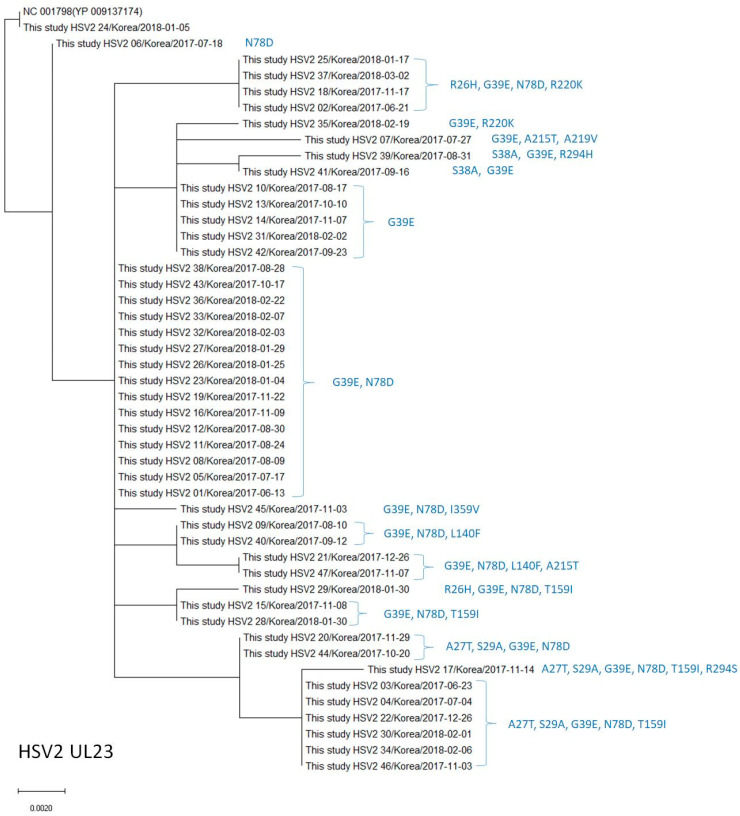
Phylogenetic analysis of the HSV-2 *UL23* amino acid sequences in this study and the reference sequence. Amino acid substitutions different from the reference sequence are indicated on the right, next to the strain name.

**Figure 4 viruses-15-01709-f004:**
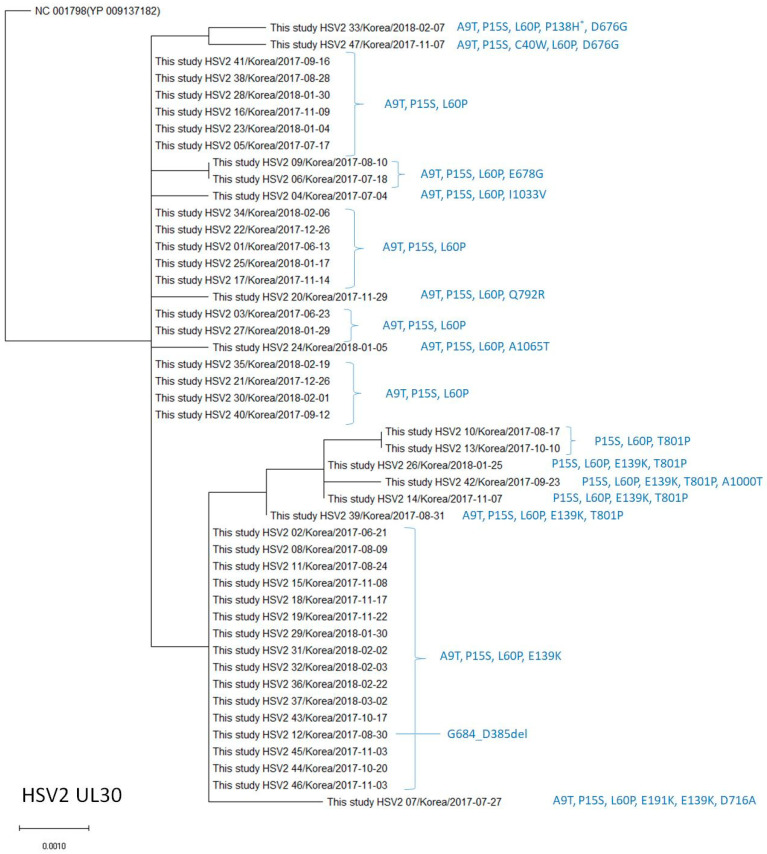
Phylogenetic analysis of the HSV-2 *UL30* amino acid sequences in this study and the reference sequence. * Amino acid substitutions different from the reference sequence are indicated on the right, next to the strain name.

**Table 1 viruses-15-01709-t001:** Clinical characteristics of the patients who were HSV-positive.

	HSV-1 (*n* = 81)	HSV-2 (*n* = 47)
Median age (range)	35 (1–88)	50 (20–82)
Sex		
Male	30 (37.0%)	17 (36.2%)
Female	51 (63.0%)	30 (63.8%)
Species of samples		
Cutaneous	58	27
Oro and nasopharyngeal	15	0
Ocular	4	0
Genital	3	15
Bronchoalveolar lavage	1	0
CSF	0	4
Pleural fluid	0	1
Immunosuppression		
HIV infection	0	0
Immunosuppressive treatment	1	3
Malignancy	0	2
Transplantation	0	0
Undescribed	80	42

CSF, cerebrospinal fluid; HSV, herpes simplex virus; HIV, human immunodeficiency virus.

**Table 2 viruses-15-01709-t002:** Primers for the amplification and sequencing of the thymidine kinase (*UL23*) and DNA polymerase (*UL30*) genes of HSV-1 and HSV-2.

Virus	Region	Primer Name	Binding	5′→3′ Sequence	* Nucleotide No. of Complete HSV Genome	Amplicon (bp)
HSV-1	TK	primer1	Forward	TTTTATTCTGTCCTTTTATTGCCGTCA	46,608	1301
		primer8	Reverse	CGAGCGACCCTGCAGCGACCCGCT	47,908	
	Pol 1	primer17	Forward	ATCCGCCAGACAAACAAGGCCCTT	62,655	2001
		primer23	Reverse	GGCCGTCGTAGATGGTGCGGGTG	64,655	
	Pol 2	primer24	Forward	GAAGGACCTGAGCTATCGCGACATC	64,462	2233
		primer30	Reverse	GGCTCATAGACCGGATGCTCAC	66,694	
HSV-2	TK	Primer9	Forward	TTTTATTCTGTTCTTTTATTGCCGTCA	46,844	1301
		Primer16	Reverse	CGAGCGACCCTGCAGCGACCCGCT	46,144	
	Pol 1	primer31	Forward	CCCGGGCGCGGGTCCGCCGGTCCG	63,159	1046
		primer34	Reverse	GTGGTGGCGTCGACGCCCCCTCG	64,204	
	Pol 2	primer35	Forward	GTGCGAAGCGGGCGCGCGCTGGCC	64,119	1042
		primer38	Reverse	GGATCTGCTGGCCGTCGTAGATGG	65,160	
	Pol 3	primer39	Forward	GGATCTGAGCTACCGCGACATC	64,961	2242
		primer44	Reverse	GGCTCATCGATCGGATGCTGAC	67,202	

HSV, herpes simplex virus; pol, polymerase; TK, thymidine kinase. * NCBI reference sequences: NC_001806.2 for HSV-1 and NC_001798.2 for HSV-2.

**Table 3 viruses-15-01709-t003:** Variant frequencies within the thymidine kinase and DNA polymerase genes among the HSV-1 and HSV-2 clinical isolates.

	HSV-1 (*n* = 81)		HSV-2 (*n* = 47)	
	TK (*UL23*)	DNA Polymerase (*UL30*)	TK (*UL23*)	DNA Polymerase (*UL30*)
No. of nucleotides	1128	3705	1128	3720
Nucleotide identity%	99.1 (98.9–99.3)	99.6 (99.5–99.7)	99.7 (99.5–100)	99.8 (99.7–99.9)
Nucleotide substitution (no.)	9.8 (8–12)	13.3 (11–17)	3.1 (0–6)	6.0 (3–11)
No. of amino acids	376	1235	376	1240
Amino acid identity (%)	98.7 (98.1–99.2)	99.5 (99.4–99.7)	99.2 (98.4–100)	99.7 (99.5–99.8)
Amino acid changes (no.)	4.9 (3–7)	5.8 (4–8)	2.8 (0–6)	3.8 (3–6)
Silent mutations (%)	50.19	56.44	9.52	37.01

HSV, herpes simplex virus; TK, thymidine kinase.

**Table 4 viruses-15-01709-t004:** Variant frequencies of the thymidine kinase (*UL23*) gene of HSV-1 in comparison with previous studies.

Thymidine Kinase (*UL23*)	This Study (*n* = 81, 2017–2018, Korea)	Burrel (*n* = 44, 2010, France) [6]	Bone (*n* = 56, 2011, Germany) [7]	Schmidt (*n* = 17, 2015, Germany) [8]	*p* Value
AA change	No. of isolates (%)				
Y4H	1 (1.2)				
C6G ^†^	49 (60.5)	14 (31.8) *	6 (10.7) *	4 (23.5) *	<0.0001
C6V	1 (1.2)				
N23S ^†^	80 (98.8)	44(100)	54 (96.4)	15 (88.2) *	0.0496
Q34P	1 (1.2)				
K36E ^†^	81 (100)	44 (100)	33 (58.9) *	15 (88.2)	<0.0001
R41H ^†^	1 (1.2)	1 (2.3)	1 (1.8)		
L42P ^†^	52 (64.2)	15 (34.1) *	11 (19.6) *	4 (23.5) *	<0.0001
D76E	1 (1.2)				
R89Q ^†^	28 (34.6)	26 (59.1) *	11 (19.6)	7 (41.2)	0.0008
E95Q	1 (1.2)				
E146G ^†^	1 (1.2)	1 (2.3)			
E257K ^‡^	2(2.5)			1 (5.9)	
A265T ^†^	80 (98.8)	43 (97.7)	56 (100)	7 (41.2) *	<0.0001
S276R ^†^	1 (1.2)		1 (1.8)	1 (5.9)	
G279D	1 (1.2)				
A294V ^†^	1 (1.2)				
D303N ^†^	1 (1.2)				
R320C	1 (1.2)				
D330N	1 (1.2)				
G335S	1 (1.2)				
V348I ^†^	8 (9.9)	9 (20.5)	7 (12.5)		
I361L	1 (1.2)				

Abbreviations: AA, amino acid; HSV, herpes simplex virus. * *p* value <0.05 between this study and other studies. ^†^ Known natural polymorphisms [3]. ^‡^ Known acyclovir resistance-related non-synonymous mutations [3].

**Table 5 viruses-15-01709-t005:** Variant frequencies of the DNA polymerase (*UL30*) gene of HSV-1 in comparison with previous studies.

DNA Polymerase (*UL30*)	This Study (n = 81, 2017–2018, Korea)	Burrel (*n* = 44, 2010, France) [6]	Bone (*n* = 56, 2011, Germany) [7]	Schmidt (*n* = 17, 2015, Germany) [8]	*p* Value
AA change	No. of isolates (%)				
P8T	1 (1.2)				
A25V ^†^	1 (1.2)		1 (1.8)		
P29H ^†^	1 (1.2)		1 (1.8)		
S33G ^†^	81 (100)	40 (90.9) *	46 (86.7) *	16 (94.1)	0.0059
A92T	1 (1.2)				
D110A	1 (1.2)				
R112H ^†^	3 (3.7)		1 (1.8)		
P137T	1 (1.2)				
I182V	4 (4.9)				
E249K	1 (1.2)				
E254A	1 (1.2)				
R264H ^†^	1 (1.2)				
V278A	1 (1.2)				
E353K	1 (1.2)				
A405V	1 (1.2)				
R663W	3 (3.7)				
D672N ^†^	47 (58.0)	3 (6.8) *	14 (26.4) *	1 (5.9) *	<0.0001
G749D ^†^	3 (3.7)	1 (2.3)	1 (1.8)		
D871N ^†^	1 (1.2)				
P875T	1 (1.2)				
V905M ^†^	78 (96.3)	33 (75) *	43 (81.1) *	15 (88.2)	0.0019
K908Q	3 (3.7)				
P920S ^†^	57 (70.4)	4 (9.1) *	11 (20.8) *	1 (5.9) *	<0.0001
A1098T	3 (3.7)				
A1099T ^†^	4 (4.9)	1 (2.3)			
E1104K ^†^	1 (1.2)			1 (5.9)	
S1113C ^†^	4 (4.9)	1 (2.3)			
E1120K	1 (1.2)				
P1124H ^†^	76 (93.8)	17 (38.6) *	25 (47.2) *	8 (47.1) *	<0.0001
P1198H	1 (1.2)				
P1198L	1 (1.2)				
P1199Q ^†^	5 (6.2)	2 (4.5)	7 (13.2)	2 (11.8)	0.4041
T1208A ^†^	80 (98.8)	37 (84.1) *	38 (71.7) *	13 (76.5) *	<0.0001
T1208V	1 (1.2)				

AA, amino acid; HSV, herpes simplex virus. * *p* value <0.05 between this study and other studies. ^†^ Known natural polymorphisms [3].

**Table 6 viruses-15-01709-t006:** Variant frequencies of the thymidine kinase (*UL23*) gene of HSV-2 in comparison with previous studies.

Thymidine Kinase (*UL23*)	This Study (*n* = 47, 2017–2018, Korea)	Burrel (*n* = 54, 2010, France) [6]	Bone (*n* = 12, 2011, Germany) [7]	*p* Value
AA change	No. of isolates (%)			
R26H ^†^	5 (10.6)			
A27T ^†^	9 (19.1)	2 (3.7) *		
S29A ^†^	9 (19.1)			
S38A	2 (4.3)			
G39E ^†^	45 (95.7)	40 (74.1)	11 (91.7)	0.1612
N78D ^†^	37 (78.7)	19 (35.2) *	9 (75.0)	<0.0001
L140F ^†^	4 (8.5)	16 (29.6) *	7 (58.3) *	0.0006
T159I ^†^	10 (21.3)			
A215T ^†^	3 (6.4)	2 (3.7)	2 (16.7)	0.2412
A219V	1 (2.1)			
R220K ^†^	5 (10.6)		1 (8.3)	
R294H	1 (2.1)			
R294S	1 (2.1)			
I359V	1 (2.1)			

AA, amino acid; HSV, herpes simplex virus. * *p* value <0.05, between this study and other studies. ^†^ Known natural polymorphisms [3].

**Table 7 viruses-15-01709-t007:** Variant frequencies of the DNA polymerase (*UL30*) gene of HSV-2 in comparison with previous studies.

DNA Polymerase (*UL30*)	This Study (*n* = 47, 2017–2018, Korea)	Burrel (*n* = 54, 2010, France) [6]	Bone (*n* = 12, 2011, Germany) [7]	*p* Value
AA change	No. of isolates (%)			
A9T ^†^	42 (89.4)	49 (90.7)	12 (100)	0.5059
P15S ^†^	47 (100)	49 (90.7)	12 (100)	0.0574
C40W ^†^	1 (2.1)		1 (8.3)	
L60P ^†^	47 (100)	49 (90.7)	12 (100)	0.0574
P138H	1 (2.1)			
E139K ^†^	21 (44.7)	16 (29.6)	2 (16.7)	0.1102
E191K	1 (2.1)			
D676G ^†^	2 (4.3)	4 (7.4)	2 (16.7)	0.3197
E678G ^†^	2 (4.3)	11 (20.4)		
G684_D685del	1 (2.1)			
D716A	1 (2.1)			
Q792R	1 (2.1)			
T801P ^†^	6 (12.8)			
A1000T ^†^	1 (2.1)	1 (1.9)		
I1033V	1 (2.1)			
A1065T	1 (2.1)			

Abbreviations: AA, amino acid; HSV, herpes simplex virus. ^†^ Known natural polymorphisms [3].

## Data Availability

The deduced amino acid sequences of the TK and pol genes of HSV-1 and HSV-2 in this study are available in Appendix A.

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
