# Peer review of "Variant Analysis of the Thymidine Kinase and DNA Polymerase Genes of Herpes Simplex Virus in Korea: Frequency of Acyclovir Resistance Mutations"

_viruses, 2023, doi:10.3390/v15081709_

Round 1
Reviewer 1 Report
The manuscript by Hyun et al. investigated the polymorphisms of UL23 and UL30 in HSV-1 and HSV-2 clinical isolates from patients in Korea. The authors have identified known natural polymorphisms in these genes, as well as several novel polymorphisms that have not been reported in the literature. The manuscript is well written and results are clearly defined and explained.
Minor concerns:
Lines 50 – 53: Are these statistics on ACV resistance for HSV-1 or HSV-2? The authors should specify here.
Line 78 – 80: PCR details are vague. Information should be provided on cycling conditions and kits used.
Discussion: the discussion is good but it lacks detail on the significance of this work. A further explanation on why this is important, and how this can lead to combating anti-viral resistance would strengthen the discussion
Reviewer 2 Report
In this manuscript, the authors sought to characterize polymorphisms of both HSV-1 and HSV-2 of a Korean cohort. Two genes, UL23 and UL30, which are actively associated with resistance to the main drugs inhibiting viral DNA replication, were chosen. Several mutations were identified, however it is unknown if the novel ones confer antiviral resistance. The methodology is fairly standard and a strength of the manuscript is the cohort size and diversity. There should be some cautionary note about the variants sequenced at around 1% of the population, as Ion Torrent has a roughly 1% error rate. Without Sanger sequencing or another technique to verify these, they may be in the expected range of technical mutations unless each sample was processed multiple times. How were the phylogenetic trees constructed? Whether the patients have been on antivirals should be described. Can the mutations be attributed to antiviral usage among the patients or are they naturally arising? The accession numbers mentioned later in the text should also be given in the description of table 2. et al should be changed to et al. and italicised Line 20: hyphenate to HSV-infected Line 24: hyphenate to TK-associated Line 41: change to kinases Line 108: change to NCBIAuthor Response
Please see the attachment.

Reviewer 3 Report
In this study, Hyun et al. applied the state-of-art next-generation sequencing to comprehensively investigate the genetic polymorphisms and their frequencies of the HSV thymidine kinase (TK) and HSV DNA polymerase (pol) genes. The meticulous analysis was conducted on a total of 81 HSV-1 and 47 HSV-2 clinical isolates from Korean HSV-infected patients. The study successfully unveiled both previously known and novel polymorphisms in TK and pol genes. The E257K amino acid substitution in TK conferring Acyclovir (ACV)-resistant was detected in 2.5% of HSV-1 isolates. Conversely, no mutation related to ACV-resistance were detected among HSV-2 clinical isolates. The frequencies of certain known polymorphisms exhibited significant variation from those previously reported in other countries. Such disparities emphasize the importance of considering regional genetic diversity and epidemiological factors when scrutinizing antiviral resistance.
An important limitation of this study, as acknowledged by the authors in the Discussion section, pertains to the absence of phenotypic test to investigate whether some of the novel polymorphisms identified herein are indeed associated with antiviral resistance. While I firmly believe in the potential significance of establishing such associations, it is essential to note that the lack of this specific information does not in any way diminish the overall importance of this study and its substantial contributions to the field of antiviral research.
In essence, the findings of this comprehensive investigation provide a remarkable contribution to the understanding of HSV genetic variation linked to antiviral resistance-related mutations. This study provides valuable information for further research in this area.
I highly recommend this manuscript for publication due to its significant findings, rigorous methodology, and contribution to the field.
To enhance the impact of their findings, I suggest that the authors refine their writing style to more effectively convey the significance of their research.
Author Response
Response to Reviewer 3 Comments
In this study, Hyun et al. applied the state-of-art next-generation sequencing to comprehensively investigate the genetic polymorphisms and their frequencies of the HSV thymidine kinase (TK) and HSV DNA polymerase (pol) genes. The meticulous analysis was conducted on a total of 81 HSV-1 and 47 HSV-2 clinical isolates from Korean HSV-infected patients. The study successfully unveiled both previously known and novel polymorphisms in TK and pol genes. The E257K amino acid substitution in TK conferring Acyclovir (ACV)-resistant was detected in 2.5% of HSV-1 isolates. Conversely, no mutation related to ACV-resistance were detected among HSV-2 clinical isolates. The frequencies of certain known polymorphisms exhibited significant variation from those previously reported in other countries. Such disparities emphasize the importance of considering regional genetic diversity and epidemiological factors when scrutinizing antiviral resistance.
An important limitation of this study, as acknowledged by the authors in the Discussion section, pertains to the absence of phenotypic test to investigate whether some of the novel polymorphisms identified herein are indeed associated with antiviral resistance. While I firmly believe in the potential significance of establishing such associations, it is essential to note that the lack of this specific information does not in any way diminish the overall importance of this study and its substantial contributions to the field of antiviral research.
In essence, the findings of this comprehensive investigation provide a remarkable contribution to the understanding of HSV genetic variation linked to antiviral resistance-related mutations. This study provides valuable information for further research in this area.
I highly recommend this manuscript for publication due to its significant findings, rigorous methodology, and contribution to the field.
To enhance the impact of their findings, I suggest that the authors refine their writing style to more effectively convey the significance of their research.
Response: We would like to thank Reviewer 3 for their time and effort in reviewing our manuscript and for providing comments. We received an English proofreading service from "Editage".
We are grateful for your positive feedback.